# Knowledge Rumination for Pre-trained Language Models

**Yunzhi Yao**[1,2], **Peng Wang**[1,2], **Shengyu Mao**[1,2], **Chuanqi Tan**[4],
**Fei Huang**[4], **Huajun Chen**[1,2,3], **Ningyu Zhang**[1,2,*]

[1] Zhejiang University
[2] Zhejiang University - Ant Group Joint Laboratory of Knowledge Graph
[3] Donghai Laboratory [4] Alibaba Group

{yyztodd,peng2001,shengyu,huajun_sir,zhangningyu}@zju.edu.cn
{chuanqi.tcq,f.huang}@alibaba-inc.com

## Abstract

Previous studies have revealed that vanilla pre-trained language models (PLMs) lack the capacity to handle knowledge-intensive NLP tasks alone; thus, several works have attempted to integrate external knowledge into PLMs. However, despite the promising outcome, we empirically observe that PLMs may have already encoded rich knowledge in their pre-trained parameters but fail to fully utilize them when applying them to knowledge-intensive tasks. In this paper, we propose a new paradigm dubbed **Knowledge Rumination** to help the pre-trained language model utilize that related latent knowledge without retrieving it from the external corpus. By simply adding a prompt like *"As far as I know"* to the PLMs, we try to review related latent knowledge and inject them back into the model for knowledge consolidation. We apply the proposed knowledge rumination to various language models, including RoBERTa, DeBERTa, and GPT-3. Experimental results on six commonsense reasoning tasks and GLUE benchmarks demonstrate the effectiveness of our proposed approach, which proves that the knowledge stored in PLMs can be better exploited to enhance performance[1].

## 1 Introduction

Pre-trained language models (PLMs) have waved the NLP community as fundamental infrastructure by demonstrating remarkable abilities with the "pre-train, prompt, and predict" paradigm (Liu et al., 2023b; Zhao et al., 2023). The mere PLMs, however, lack the capacity to handle knowledge-intensive tasks with advanced functionalities like commonsense reasoning (Lin et al., 2019; Qiao et al., 2022; Liu et al., 2023a) and open-domain question answering (Yang et al., 2015). This necessitates a boosting trend for research focusing

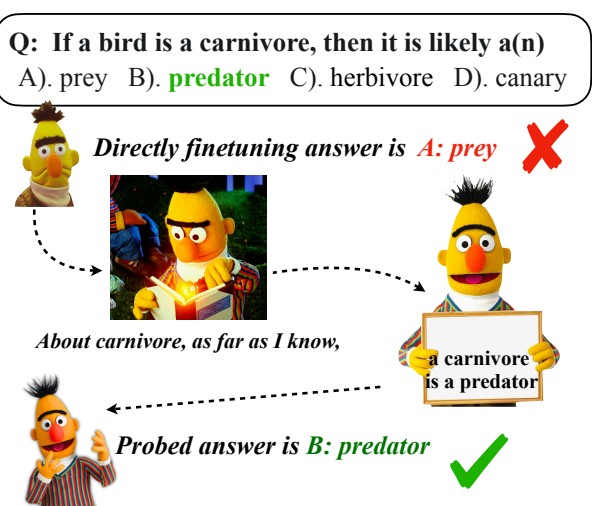

**Q: If a bird is a carnivore, then it is likely a(n)**
A). prey   B). **predator**   C). herbivore   D). canary

*Directly finetuning answer is A: prey* ❌

*About carnivore, as far as I know,*

*a carnivore is a predator*

*Probed answer is B: predator* ✅

Figure 1: A pilot experimental case for the motivation of knowledge rumination. The PLM succeeds in probing the related commonsense knowledge but fails to obtain the answer with finetuning.

on augmenting PLMs with external knowledge sources (Chen et al., 2017, 2022a; Welleck et al., 2021, 2022; Zhang et al., 2022, 2023).

However, despite the empirical success, we observe that PLMs can often encode extensive knowledge within their parameters yet fail to utilize this effectively for knowledge-intensive tasks. Taking pilot experiments as an example, we use knowledge probing (Petroni et al., 2019) to the PLM as shown in Figure 1. Given a question *"If a bird is a carnivore, then it is likely a(n) what?"*, we notice that the PLM has known the knowledge *"a carnivore is likely a(n) predator"* in its parameters; however, we surprisingly find that the finetuned PLM chose the wrong answer despite the model knowing the related knowledge. Interestingly, this phenomenon mirrors human behavior. As an example, in the cognitive reflection test (CRT) (Frederick, 2005), participants have posed a series of straightforward questions (already learned), yet they often initially fail in their intuitive reason-

---

*Corresponding author.
[1]Code is available in https://github.com/zjunlp/knowledge-rumination.

ing. Upon reflection, however, individuals typically identify their erroneous responses and correct them. Consequently, we conjecture that the prominent PLMs of today have flaws as humans and we still have the following problem: *are we fully exploiting the potential of the PLMs?*

Some pioneering researchers have attempted to unravel this enigma. For instance, Chen et al. (2022b) and van de Kar et al. (2022) propose to utilize the knowledge in the pre-traning corpus by retrieve-then-fine-tuning method. Likewise, Bhagavatula et al. (2020) capitalizes on the implicit knowledge within large language models (>10B) by retrieving from model weights with recitation-augmented generation. These studies affirm that PLMs encapsulate a vast body of knowledge, with untapped potential, while in our paper, we pursue a more universally applicable, yet simple solution to fully harness knowledge in PLMs for NLP.

To address this need, we introduce **Knowledge Rumination** to assist the model in thinking thoughtfully in handling knowledge-intensive tasks. Analogous to how animals ruminate food for better digestion and absorptionby regurgitating it from the stomach back to the mouth for additional chewingwe aim to mimic this process by having the model first review the relevant knowledge stored in its parameters and then consolidate this knowledge to better tackle associated tasks. In detail, we propose knowledge reviewing with a task-guided prompt by simply adding *"As far as I know"* to stimulate the model to recall latent knowledge. Subsequently, we consolidate knowledge via FFN to explicitly leverage latent knowledge to help address downstream tasks since FFN plays a crucial role in PLMs (Wang et al., 2022).

We apply the proposed knowledge rumination to various PLMs, including RoBERTa (Liu et al., 2019), DeBERTa (He et al., 2021). We also transfer knowledge rumination to large language GPT-3 (175B) (Brown et al., 2020). Experimental results on six commonsense reasoning tasks and the GLUE benchmark demonstrate that the proposed simple method can obtain performance gain and even outperform baselines of retrieving external knowledge. To conclude, we summarize the contributions of this work as follows:

- We propose a novel approach of **Knowledge Rumination** to better utilize the knowledge stored in the parameters, which is model agnostic and can be applied to any PLMs

- Experimental results demonstrate that the proposed approach can successfully elicit related knowledge for both small and large PLMs, yielding better performance on six commonsense tasks and GLUE benchmarks.

- Comprehensive empirical analysis indicates that still a large underestimated amount of knowledge can be retrieved from PLM's model weights, and our work takes a small step in this direction.

## 2 Related Work and Background

**Extracting Knowledge from PLMs** Previous studies have shown that PLMs implicitly contain a large amount of knowledge. Petroni et al. (2019) have shown that such language models can be used in a Knowledge Base (KB) completion task by converting KB relations into natural language templates. Based on this finding, researchers attempt to treat the PLM as a knowledge base. Some studies (Bosselut et al., 2019; West et al., 2022; Hao et al., 2022) employ PLMs to construct knowledge graphs automatically. Meanwhile, some others (Shwartz et al., 2020; Li et al., 2022) find that the knowledge possessed by the PLMs can be used to enhance the model's performance in downstream tasks. To date, several work (Wang et al., 2023; Zelikman et al., 2022; Bhagavatula et al., 2020) attempt to utilize PLMs to generate free-text rationales for reasoning. Our approach differs from previous works in that we aim to enhance the model's understanding of what it already knows in order to improve performance.

**Knowledge-Enhanced Models** Researchers resort to external sources to facilitate the model's ability to deal with knowledge-intensive situations. One direction is to ground the question in a KB and conduct inference with both the question and the retrieved knowledge (Yasunaga et al., 2022, 2021; Zhang et al., 2022; Sun et al., 2019; Yao et al., 2022; Lv et al., 2020; Lin et al., 2019). Since the pre-trained model can also be viewed as a knowledge store, several recent studies including Self-talk (Shwartz et al., 2020), Rainier (Liu et al., 2022a), GKP (Liu et al., 2022b), ElicitKnowledge (Li et al., 2022) propose to treat the large language model (e.g., GPT-3) as an external source to elicit knowledge for downstream tasks. In contrast, our approach diverges from relying on external sources such as knowledge bases (KB)

or language models (LM). Instead, we concentrate on fully leveraging the latent knowledge acquired by the model itself. There are also some kinds of work that decompose the question into sub-questions and ask the model to answer each sub-question such as least-to-most prompt (Zhou et al., 2023). However, even these approaches encounter the issue we proposed where the model may possess the answer to the sub-question within its parameters but fails to provide the correct response. The underlying intuition behind our method is that current methods for harnessing the power of pre-trained language models (PLMs) have not fully tapped into the knowledge residing within the model's parameters.

Note that our approach most closely aligns with Self-talk (Shwartz et al., 2020), but with an additional capability to manage parametric knowledge (such as embeddings in Feed-Forward Networks). This capability broadens the spectrum of the academic idea to a certain extent.

## 3 Knowledge Rumination

In this section, we introduce technical details of **knowledge rumination** to tap into the potential of PLMs (§3.1). Given a PLM $G$, we first freeze the model parameters and design a task-specific prompt (§3.2) to guide the model in reviewing its stored knowledge regarding the task and input (knowledge reviewing). We then consolidate the model's latent knowledge (§3.3) during tuning downstream tasks (knowledge consolidation).

### 3.1 Model Architecture

We take a representative task, multiple-choice commonsense reasoning, as an example to elucidate the details of knowledge rumination, which can be simply adapted to any other tasks in NLP. Given a question $q$, multiple-choice commonsense reasoning aims to selecting the correct answer $a_k \in \mathcal{A}$ provided with an optional context $c$. The set of possible answers $\mathcal{A}$ is finite and varies for each question. In the vanilla setting, the PLM is used to directly answer the question by selecting the answer choice $\hat{a}$ with the highest score, based on the concatenation of the question $q$, context $c$, and one possible answer choice $a_i$ as:

$$\hat{a} = \arg\max_{a_i \in \mathcal{A}} P(a_i \mid c, q) \quad (1)$$

Here, before making a prediction, we ask the model to carefully consider the question and re-

view its prior knowledge. We freeze the PLM $G_\theta$ to probe the knowledge it has stored ($\theta$ represents the model's parameter) and prepend trainable continuous tokens to each layer. For each question $q$, we create a unique prompt $p_q$ to guide the model in reflecting on its knowledge:

$$r = G_\theta([q; p_q]) \quad (2)$$

Then, the PLM will reinforce its knowledge $r$ of the problem and infer the answer augmented with $r$. Ideally, the model is supposed to generate helpful knowledge *texts* for the question. However, training the model requires expensive knowledge annotations for all training instances. To handle this problem, we use the model's *contextualized representation output as the knowledge* for rumination and leverage it as a latent variable. The model will answer the question based on both the question $q$ and the vectorized knowledge $r$:

$$\hat{a} = \arg\max_{a_i \in \mathcal{A}} P(a_i \mid c, q, r) \quad (3)$$

Then, the cross-entropy loss is used to train the whole model. Assuming the answer $a_k$ is correct, the loss can be obtained as follows:

$$\mathcal{L}_{ce} = -\sum_{a_i \in \mathcal{A}} Q(a_i \mid c, q) \log P(a_i \mid c, q, r) \quad (4)$$

where $Q(a_i \mid c, q)$ is 1 if $a_i = a_k$ and 0 otherwise. During training, the gradient flows back into the model, which helps it learn to review and consolidate useful information.

### 3.2 Knowledge Reviewing with Task-guided Prompting

Analogically, animals return partially digested food from the stomach to the mouth for re-chewing; we design specific prompts for each question to probe the latent knowledge for rumination. As shown in Figure 2, we begin by using the background prompt: *"As far as I know, [MASK]"*. Note that humans consider mentions in the descriptions to better understand the question. For example, when answering the question *" If a bird is a carnivore, then it is likely a(n) what?"*, humans would consider the mentions of *bird* and *carnivore* to better comprehend the question. We further introduce the mention prompt to review knowledge of mentions. Specifically, we extract mentions $M$ from the questions using off-the-shelf tools[2] and

---

[2] https://github.com/marcocor/tagme-python

**(a) Fine-Tuning for PLM**

CLS Head ----→ Answer: C

[CLS] Q: Where would I … ? Choice: hen house. [SEP]

**(b) Prompt Learning for PLM**

great(positive)
terrible (negative) ◄---- Verbalizer

[CLS] No reason to watch. It was [MASK] [SEP]

**(c) Knowledge Rumination for PLM**

Contextualized Embeddings

Rumination Distribution

england
mountains
hen house
english hunt
california

Answer: C

*FFN*
*value* $\phi_k$
activation
*key* $\phi_v$

**Knowledge Consolidation**

proj ← $\mathbf{h}_{[MASK]}$ $r$

$\theta$ · · · · *Prefix* ❄ *Frozen PLM*

[CLS] Where would I not want a fox?

As far as I know, [MASK] … [SEP]
About fox, I know, [MASK] … [SEP]

Figure 2: An illustration of different methodologies to utilize the PLM. (a): standard finetuning, (b): prompt learning, and (c) the proposed knowledge rumination method. During the knowledge reviewing with task-guided prompting (§3.2), the model parameters are frozen. $\mathbf{h}_{[MASK]}$ (the hidden vector of "[MASK]") is the elicited latent knowledge, which will be injected into FFNs for knowledge consolidation (§3.3).

build prompts to elicit memories about these mentions. However, it should be noted that some mentions contain unrelated information and may divert attention. To address this, we propose "mention relevance scoring," where we utilize an encoder model $G_{\text{enc}}$ to evaluate the relevance of each mention in relation to the question context. Specifically, we compute the relevance score for each mention $m \in M$ by concatenating the text with the question $q$ and using the output of "[CLS]" as the score ($f_{\text{cls}}$ in the following equation):

$$\rho_m = f_{\text{cls}}\left(G_{\text{enc}}([q; m])\right) \quad (5)$$

We sample mentions with the Top-2 relevance scores $\rho$ as the target mentions. The details of "mention relevance scoring" can be found in Appendix A.5. Actually, apart from this commonsense or context knowledge, it is important to note that PLMs also store other types of knowledge, including skill knowledge and task knowledge (Wang et al., 2022). Hence, we assume that the model may acquire latent knowledge regarding the task itself, so we further build task prompts to encourage the model to reflect on the skill knowledge encoded within its parameters.

Overall, we probe the PLM using three different types of task-guided prompts, along with three areas of interest:

- **Background Prompt**: is designed to help the model to think about the background, the prompt is *As far as I know [MASK].*

- **Mention Prompt**: is used to elicit memories of mentions, the formation is *About <Mention>, I know [MASK].*

- **Task Prompt**: is designed to help the model reminisce memories of the task. For example, for the sentiment analysis, the prompt is *About sentiment analysis, I know [MASK].*

We put several *'[MASK]'* in the task-guided prompt, and the length of the *'[MASK]'* is a *hyperparameter* for different tasks.

Our approach can be applied to different PLMs. For the encoder-style model like RoBERTa, we utilize the hidden states $\mathbf{h}_{[MASK]}$ of the *'[MASK]'* as the latent knowledge for rumination. $f_{\text{mask}}$ in the following equation means taking the $\mathbf{h}_{[MASK]}$ from the model's output.

$$r = f_{\text{mask}}(G_\theta([q; p_q])) \quad (6)$$

The following section will explain how to utilize this elicited knowledge.

|  |  | CSQA | SocialIQA | aNLI | OBQA | PIQA | HellaSwag |
|---|---|---|---|---|---|---|---|
| **Basic Model** | **RoBERTa*** | 68.7 | 75.9 | 82.7 | 64.9 | 79.4 | 82.3 |
|  | **DeBERTa** | 72.4 | 77.2 | 86.0 | 74.6 | 81.1 | 89.0 |
| **Ext. Knowledge** | **QAGNN*** | 73.4 | 75.7 | 83.0 | 67.8 | 79.6 | 82.6 |
|  | **GreaseLM*** | 74.2 | 75.5 | 83.3 | 66.9 | 79.6 | 82.8 |
|  | **Dragon*** | **76.0** | 76.8 | 84.0 | 72.0 | 81.1 | 85.2 |
| **Rumination Model** | **RumiRoBERTa** | 70.3 | 77.7 | 86.1 | 70.0 | 80.8 | 85.8 |
|  | **RumiDeBERTa** | 74.3 | **78.4** | **86.7** | 76.0 | **81.9** | **89.5** |

Table 1: Accuracy on downstream commonsense reasoning tasks. Scores of the methods marked with * are taken from Yasunaga et al. (2022). As the official tests for CSQA, PIOA and HellaSwag are hidden, here we report the in-house Dev (IHdev) and Test (IHtest) accuracy, following the data split in Yasunaga et al. (2022).

## 3.3 Knowledge Consolidation with FFN Neuron Augmentation

To reinforce its understanding, the model should re-digest (inject) the elicited knowledge $r$ of the $q$, similar to how animals chew their food again. However, where to inject the PLMs remains a challenging issue, indicating potential work to investigate how the model's knowledge is kept. Previous studies (Dai et al., 2022; Wang et al., 2022) have discovered that the Feed Forward Network works (FFN) as the knowledge neuron or skill neuron, illustrating that FFN may store factual information and encode task-specific skills. Inspired by these findings, we incorporate $r$ into the FFNs, as previous work (Yao et al., 2022) does. Here, we select the Top-1 layer to re-diest (inject) the knowledge. Suppose the two linear layers in FFN emulate as a key-value network $K$ and $V$, we employ two distinct linear layers to project the information $r$ to the vector space of the matching layer:

$$\phi_k = W_k \cdot r \qquad (7)$$
$$\phi_v = W_v \cdot r \qquad (8)$$

where $W_k$ and $W_v$ represents the weights of the two linear layers ($W_k, W_v \in \mathbb{R}^{d \times d}$, $d$ is the intermediate size of the PLM). The two matrices, $W_k$ and $W_v$, are initialized randomly and will be updated during training. We expand the FFN by concatenating the projected knowledge to the end of the linear layer and obtain the expanded $K_E, V_E$. The computing can be described as follows:

$$\begin{aligned} FFN(H) &= f(H \cdot K_E) \cdot V_E \\ &= f(H \cdot [\phi_k : K]) \cdot [\phi_v : V] \end{aligned} \qquad (9)$$

$H$ denotes the output hidden states of the self-attention module. The model would answer the question with the help of regurgitated knowledge.

## 4 Experiments

### 4.1 Dataset

We evaluate the proposed approach on six knowledge-intensive tasks of commonsense reasoning benchmarks: CommonsenseQA (**CSQA**) (Talmor et al., 2019), SocialIQA(Sap et al., 2019), PhysicalQA (**PIQA**) (Bisk et al., 2020), Openbook QA (**OBQA**) (Mihaylov et al., 2018), HellaSwag (Zellers et al., 2019) and Abductive Natural Language Inference (**aNLI**) (Bhagavatula et al., 2020). We follow the data split used by prior works (Yasunaga et al., 2022). Meanwhile, to better understand the effect of the task prompt in our method and the skill knowledge (Wang et al., 2022) learned in the pre-trained language model, we consider tasks of the GLUE benchmarks (Wang et al., 2018), including single-sentence tasks (**SST, CoLA**), inference tasks (**QNLI, RTE**), and similarity and paraphrase tasks (**STS-B, MRPC**). We provide dataset details in Appendix A.7.

### 4.2 Baselines

We choose RoBERTa_large (Liu et al., 2019) and DeBERTa_large (He et al., 2021) as our backbone models for moderately sized language models and compare performance with the following strong external knowledge-enhanced approaches: **QAGNN** (Yasunaga et al., 2021), **GreaseLM** (Zhang et al., 2022) and **Dragon** (Yasunaga et al., 2022). More details about baseline models can be seen in Appendix A.3.2. For the tasks in the GLUE benchmark, in addition to RoBERTa_large, we compare our model with a prompt learning method, LM-BFF (Gao et al., 2021). LM-BFF proposes a prompt-based finetuning method and a refined

|  |  | SST-2 | SST-5 | CoLA | RTE | MRPC$_{f1}$ | QNLI | STS-B$_{pear}$ | AVG |
|---|---|---|---|---|---|---|---|---|---|
| **Basic Model** | **RoBERTa** | 95.00 | 58.70 | 62.60 | 80.90 | 91.40 | 93.30 | 91.90 | 81.97 |
|  | **LM-BFF** | 95.41 | 60.67 | 69.27 | 86.28 | 92.76 | 94.60 | 92.00 | 84.42 |
| **Rumination** | **RumiRoBERTa** | 95.75 | 60.85 | 68.57 | 85.92 | **93.92** | **94.84** | **92.23** | 84.58 |
| **Model** | **RumiLM-BFF** | **96.21** | **61.17** | **70.85** | **86.64** | 93.73 | 94.64 | 92.05 | **85.04** |

Table 2: Supervised GLUE benchmark results. Here, we report the results on the validation set following LM-BFF (Gao et al., 2021). The prompt used here is the same as LM-BFF.

strategy for dynamically and selectively incorporating demonstrations into each context. In our paper, we simply use the human-curated prompt (provided by LM-BFF) and leverage RoBERTa_large as the backbone.

### 4.3 Experiment Implementation

In the stage of knowledge reviewing with task-guided prompting, the backbone of the model is frozen, and we only update the prepended trainable continuous tokens (prefix prompt). When implementing the DeBERTa, due to the complex attention mechanism, we simply freeze the whole model and do not add the prefix tokens. For the commonsense reasoning task, we combine the mentioned prompt and background prompt, and for the GLUE benchmarks, we find the task prompt to be more useful. More details can be found in Appendix A.2 and A.3.

### 4.4 Main Results

We list the results in Table 1 for the commonsense reasoning tasks and Table 2 for the GLUE benchmark. The proposed technique, called knowledge rumination, demonstrates superior performance on a majority of datasets. To be noted, it outperforms naive baselines and obtains better or comparable performance with baselines that incorporate external knowledge. As the Table illustrates, the proposed method of knowledge rumination, RumiRoBERTa, and RumiDeBERTa, shows improved performance on six commonsense reasoning tasks. The results demonstrate that RumiRoBERTa and RumiDeBERTa consistently outperform existing language models (RoBERTa and DeBERTa), with a notable improvement of +2% absolute accuracy on CSQA compared to RoBERTa and DeBERTa. These results indicate that the knowledge stored in the PLM's parameters can still be further exploited. In addition, it can be observed that RumiRoBERTa and the method that incorporates external knowledge have compa-

rable results. Notably, on SocialIQA, aNLI, and HellaSwag, RumiRoBERTa even outperforms the pre-trained, knowledge-enhanced model Dragon. On the other hand, RumiDeBERTa performs better than Dragon on most of the tasks. However, the model that uses external knowledge performs best on the CommonsenseQA task. It is hypothesized that some commonsense reasoning datasets are derived from pre-existing knowledge bases. For instance, CommonsenseQA is derived from ConceptNet. The above observations suggest that: 1) the knowledge stored in the parameters is robust and requires explicit activation during finetuning. 2) the performance of the model that retrieves knowledge from external sources is impacted by the quality and relevance of the knowledge sources, while our rumination methods can produce more pertinent knowledge.

As shown in Table 2, the results of the GLUE benchmark demonstrate that the knowledge rumination method outperforms the basic finetuning model RoBERTa and the prompt-based method LM-BFF, with an average improvement of +1% for LM-BFF and 3% for RoBERTa. These gains in performance highlight the effectiveness of knowledge rumination methods compared to finetuning and prompt learning.

### 4.5 Out-of-Distribution (OOD) Performance

To better illustrate the wide applicability and generalization prowess of the knowledge rumination method, we extended our evaluation to incorporate performance on out-of-distribution (OOD) test sets. Table 3 presents a comparative study of the OOD performance of fine-tuning techniques applied to both RoBERTa and RumiRoBERTa models. In general, RumiRoBERTa demonstrates superior performance on OOD tests compared to the conventional fine-tuning model. Notably, when RumiRoBERTa was trained on OBQA and subsequently tested on HellaSwag and PIQA, it achieved a 5% advantage over RoBERTa. This

| Method | Finetuning | RumiRoBERTa |
|---|---|---|
| **CSQA ⇒ OBQA** | 50.04 | **52.20** |
| **CSQA ⇒ SocialIQA** | 46.17 | **50.53** |
| **CSQA ⇒ HellaSwag** | **46.60** | 45.46 |
| **OBQA ⇒ PIQA** | 58.59 | **63.38** |
| **OBQA ⇒ HellaSwag** | 33.37 | **39.07** |
| **PIQA ⇒ CSQA** | 49.07 | **51.97** |
| **PIQA ⇒ HellaSwag** | 38.47 | **48.97** |
| **PIQA ⇒ OBQA** | **46.20** | 44.60 |
| **HellaSwag ⇒ CSQA** | 47.30 | **58.20** |
| **HellaSwag ⇒ PIQA** | 38.47 | **48.97** |
| **HellaSwag ⇒ SocialIQA** | 36.20 | **48.60** |

Table 3: **OOD Results.** Performance (accuracy) of the compared methods, which are firstly trained on a source dataset and then directly conduct prediction on a target dataset (denoted as source ⇒ target).

enhancement in performance can be attributed to the importance of knowledge rumination in effective problem-solving and knowledge application. Despite exhibiting slightly sub-optimal results on CSQA ⇒ HellaSwag and PIQA ⇒ OBQA tests, RumiRoBERTa's performance still compares favorably with the traditional fine-tuning method.

| | SocialIQA | OBQA | HellaSwag |
|---|---|---|---|
| **RoBERTa** | 75.9 | 64.9 | 82.3 |
| **Concat** | 77.2 | 68.8 | **85.8** |
| **FFN** | **77.7** | **70.9** | 85.7 |

Table 4: Results of different knowledge integration methods on three commonsense reasoning tasks. The backbone model is RoBERTa_Large.

# 5 Analysis

## 5.1 Impact of Different Knowledge Consolidation Methods

Apart from injecting knowledge in FFN, we compare and evaluate another injection method: Concatenation. Since the knowledge $r$ is a vector, we concatenate it into the sequence after the embedding layer and report the results in Table 4. We notice that both methods benefit from knowledge rumination. Typically, integrating the knowledge through feed-forward networks (FFN) demonstrates better performance than concatenation, with an average improvement of +0.5% in SocialIQA and +2.1% in OBQA. This supports previous research findings that Feed-Forward Networks (FFNs) store some factual knowledge (Dai et al., 2022; Yao et al., 2022) and that our method can ef-

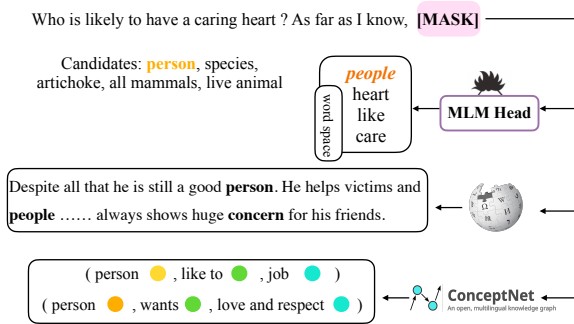

Figure 3: Case study of **Knowledge Rumination** in CommonsenseQA dataset. Given the question, we map the '[MASK]' token into vocabulary space, pre-train corpus space and external corpus space. We observe that successful knowledge rumination always contains similar information with answers.

fectively consolidate this knowledge within FFNs.

## 5.2 What does the Model Ruminate?

Despite the advantages of knowledge rumination, it is essential to understand the nature of the knowledge generated and the mechanism behind the method. In our model, the model produces a contextualized embedding $r$ as latent knowledge. To make the knowledge more interpretable to humans, we convert the vectorized knowledge to symbolic text. In order to evaluate the effectiveness of the method, we sample successful cases where the simple finetuning model makes the incorrect prediction while our knowledge rumination method provides the correct answer.

Figure 3 illustrates an example from the CommonsenseQA task. To generate the output words in the vocabulary space, we apply the masked language modeling (MLM) head over the position of the '[MASK]'. We notice that the masked word often includes similar information to the answer. For example, in the question "Who is likely to have a caring heart?", the '[MASK]' token contains words such as 'people,' 'heart,' and 'like.' In addition, we map the knowledge rumination output $r$ to the pre-trained corpus space as the memory is constructed during pre-training. We conduct a dense embedding similarity search to identify what our generated contextualized representation is most similar to. In this case, we represent the ruminated knowledge by taking the average of the '[MASK]' embedding. For each sample from external sources, we add a '[MASK]' token at the end of the sentence and use the '[MASK]' to represent the sentence. We use the pre-trained

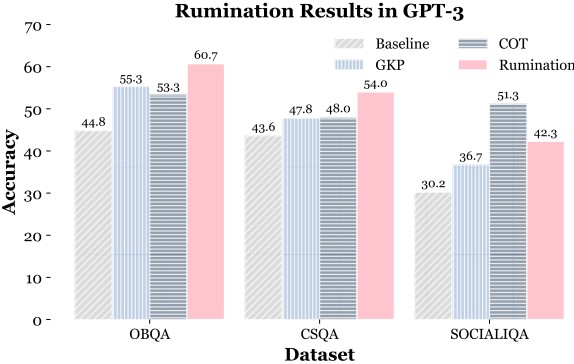

Figure 4: Results of GPT-3 on commonsense reasoning datasets. Baseline refers to GPT-3 answering the question directly with few-shot demonstrations.

corpus Wikipedia (Milne and Witten, 2008) as the retrieval source and employ FAISS (Johnson et al., 2021) for dense vector search. Interestingly, we notice that the model recalls its memory of a person with a caring heart, "a good person. He helps victims and people.". This suggests that the model has remembered this information during pre-training, and if it is given a chance to think, the model is aware of what it has learned. Besides, we also map the knowledge into the external knowledge source ConceptNet (Speer et al., 2017) since CommonsenseQA is derived from it. More details can be found in Appendix A.6.

### 5.3 Transfer to LLMs

In this part, we endeavor to transfer knowledge rumination to LLMs. Compared to small language models, LLMs demonstrate an excellent ability to recall and deal with knowledge by few-shot demonstration as a prompt. GKP (Liu et al., 2022b) notes that pre-appending knowledge retrieved from LLMs can facilitate both the LLMs and other models in effectively handling tasks. As such, we follow suit (Liu et al., 2022b,a) in our knowledge review process, using prompts to recall the memorized knowledge for the target inputs. Nevertheless, simply concatenating the recalled knowledge might not lead to effective utilization by the model. Here, in the knowledge consolidation phase, we explicitly ask the model to attentively consider the recalled knowledge by crafting a stimulus prompt "According to the [*knowledge*], the answer is" or "think by the [*knowledge*]". Furthermore, recent work suggests that Chain-Of-Thought (COT) (Wei et al., 2022) can elicit language LLMs' reasoning ability by employing a se-

ries of intermediate reasoning rationales. In contrast, the knowledge generated by the Knowledge Rumination model contains implicit information derived from pre-trained language models during pre-training which may otherwise be overlooked by the reasoning steps. Here, we report the results on GPT-3 Davinci (175B) with knowledge rumination in Figure 4 and compared with original few-shot GPT-3, GKP, and COT. The implementation details can be found in Appendix A.3.2 and the demonstrations can be found in Appendix A.2. Our findings indicate that the performance of GPT-3 can be significantly enhanced through knowledge rumination, as evidenced by the 16% improvement in OBQA accuracy, 12% in CSQA, and 11% in SocialIQA. Compared to GKP, it's evident that merely concatenating the elicited knowledge doesn't adequately leverage it. In contrast, the knowledge rumination approach surpasses GKP by an average of 6%, demonstrating its efficacy. What's more, knowledge rumination attains better performance than COT on OBQA and CSQA except for the SocialIQA, demonstrating the effectiveness of the background knowledge. In this preliminary exploration, we discovered that guiding LLMs to deliberate thoroughly on recalled knowledge can augment their understanding and reasoning capabilities. Looking forward, enhancing the utilization and integration of the model's inherent knowledge for reasoning remains a promising area for future investigation.

**Error Analysis** We conduct an error analysis on the evaluation examples from the OBQA and CSQA datasets for the GPT-3 model. We categorize the errors into four categories: 1): Failure to Utilize: the model recalls helpful information but does not provide the correct answer. 2): Ineffective Rumination: the rumination information with the highest *logprobs* is irrelevant to the question, but there are some relevant ones in the remaining. 3): Incorrect Memory: the model's stored information about the question is incorrect. 4): Missing Information: the model does not have the necessary information about the problem. Examples for each error type can be seen in Appendix A.8.

The statistics are presented in Table 5. We observe that the majority of errors are caused by missing information, indicating that large pre-trained language models still have difficulty retaining all the knowledge acquired during pre-training. Additionally, our method still has difficulty acti-

| | OBQA | CSQA |
|---|---|---|
| **Failure to Utilize** | 24% | 18% |
| **Ineffective Rumination** | 27% | 32% |
| **Incorrect Memory** | 7% | 9% |
| **Missing Information** | 42% | 41% |

Table 5: Error analysis on OBQA and CSQA.

vating all the stored knowledge, as 32% of error cases are caused by ineffective rumination and 18% by failure to utilize retrieved information for the CSQA task. This suggests that there is still room for improvement in this area.

## 6 Conclusion and Future Work

In this work, we propose knowledge rumination for PLMs, which can serve as a general solution to exploit latent knowledge for downstream tasks and demonstrate promising results. This concept is akin to humans often erring when answering without thorough thinking. In the future, we plan to apply knowledge rumination to more NLP tasks and more types of models.

## Acknowledgment

We would like to express gratitude to the anonymous reviewers for their kind comments. We thank Jiacheng Liu for his kind suggestions. This work was supported by the National Natural Science Foundation of China (No.62206246), Zhejiang Provincial Natural Science Foundation of China (No. LGG22F030011), Ningbo Natural Science Foundation (2021J190), Yongjiang Talent Introduction Programme (2021A-156-G), CCF-Baidu Open Fund, and Information Technology Center and State Key Lab of CAD&CG, Zhejiang University.

## Limitations

The proposed work still has some limitations to address in future work.

**Method.** One limitation of knowledge rumination is that it cannot handle incorrect memory as shown in §5.3, and may amplify the effects of those errors (there may exist a tug of war between task data optimization and knowledge consolidation from PLMs). Adatable retrieval-based methods may be a solution for this issue, and we leave this for future work.

**PLMs.** We apply knowledge rumination to four PLMs; however, it is still unknown whether the proposed approach works for other language models such as T5 (Raffel et al., 2019), BART (Lewis et al., 2020) and so on. We plan to extend our work in the future to cover more PLMs and multimodal, multilingual scenarios. Besides, knowledge stored in PLMs may have factual errors or severe bias, and knowledge rumination may augment such behavior.

**Tasks.** We only evaluate text classification and commonsense reasoning tasks. Due to the limited budget and computation resources, we cannot afford evaluation on more tasks. We will plan to evaluate the proposed approach on more NLP benchmark datasets such as KILT (Petroni et al., 2021).

## Ethical Considerations

The model in this paper is indented to be used for exploratory analysis of PLMs. Note that the pre-training corpus contains rich biased data; thus, the proposed knowledge rumination approach may elicit some knowledge with offensive language or discriminatory.

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

# A   Appendix

## A.1   Detailed Comparison with Previous Approaches

Specifically, we note that Self-talk (Shwartz et al., 2020), Rainier (Liu et al., 2022a), GKP (Liu et al., 2022b), and ElicitKnowledge (Li et al., 2022) all harness knowledge, in the form of text sequences, extracted from pre-trained language models to enhance performance in knowledge-intensive tasks.

Similarly, the concept of Knowledge Rumination draws from the same inspiration, enabling pre-trained language models to leverage related latent knowledge without the need for retrieval from an external corpus. Among these prior studies, our method bears the closest resemblance to Self-talk (Shwartz et al., 2020), with the added capability of Knowledge Rumination to handle parametric knowledge (e.g., embeddings in Feed-Forward Networks). This extends the scope of the academic concept to a certain degree.

Our work also shares a connection with COT (Wei et al., 2022). However, while COT generates rationales (texts) and appends them to output sequences to assist reasoning, our Knowledge Rumination model generates implicit knowledge and integrates it with the input sequence to produce desired results. Additionally, COT is primarily focused on reasoning, thus its rationales serve as intermediary steps in the reasoning process. By contrast, the knowledge generated by our Knowledge Rumination model constitutes implicit information derived from pre-trained language models during the pre-training phase.

## A.2   Prompts for Knowledge Rumination with LLM

Table 7 through Table 9 shows the full prompts for knowledge rumination that we use for each evaluated task (demonstrations are derived from Liu et al. (2022b,a)): CSQA, OBQA, and SOCIALIQA.

## A.3   Experimental Settings

In this section, we describe the implementation of our experiments in detail, including the baseline methods, backbone models, and hyperparameters. Our model is built based on the Huggingface framework (Wolf et al., 2020). Unlike finetuning, which updates all model parameters $\theta$ of a PLM, prefix-tuning freezes all pre-trained Transformer parameters and only optimizes prefix vectors that are prepended to each Transformer layer. We use prefix-tuning (Li and Liang, 2021) to train the knowledge reviewing model to reflect information for each task because: 1) the rumination models for different tasks can share the same backbone Transformer parameters, with only the prefix vectors being different. 2) Prefix-tuning has comparable performance to finetuning but avoids the risk of catastrophic forgetting.

For the tasks in the GLUE benchmarks, most of the hyperparameters are the default parameters of LM-BFF. For commonsense reasoning tasks, we follow previous preprocessing from QA-GNN (Yasunaga et al., 2021). We chose RoBERTa (Liu et al., 2019) large and DeBERTa (He et al., 2021) large as our backbone models, and the average training time for each model is 2 to 4 hours. We apply grid search for each hyperparameter tuning.

### A.3.1 Hyperparameters

The detailed hyperparameter search space is as follows: (maximum values bolded below)

*CommonsenseQA (CSQA)* .
- epoch: [5, **10**, 15]$_{roberta}$, [**5**, 10, 15]$_{deberta}$
- batch size: [8, **16**, 32]$_{roberta}$, [8, **16**, 32]$_{deberta}$
- learning rate: [5e-6, **1e-5**]
- rumi length: [7, 10, **15**]$_{roberta}$, [3, **5**, 7]$_{deberta}$

*OpenbookQA (OBQA)* .
- epoch: [**5**, 10, 15]$_{roberta}$, [**5**, 10, 15]$_{deberta}$
- batch size: [8, **16**, 32]$_{roberta}$, [8, **16**, 32]$_{deberta}$
- learning rate: [5e-6, **1e-5**]
- rumi length: [7, **10**, 15]$_{roberta}$, [3, 5, 7]$_{deberta}$

*Social Interaction QA (SocialIQA)* .
- epoch: [**5**, 10, 15]$_{roberta}$, [**5**, 10, 15]$_{deberta}$
- batch size: [8, **16**, 32]$_{roberta}$, [8, **16**, 32]$_{deberta}$
- learning rate: [**5e-6**, 1e-5]
- rumi length: [7, 10, **15**]$_{roberta}$, [3, **5**, 7]$_{deberta}$

*Physical Interaction QA (PIQA)* .
- epoch: [10, 15, **20**]$_{roberta}$, [10, 15, **20**]$_{deberta}$
- batch size: [8, 16, **32**]$_{roberta}$, [8, 16, **32**]$_{deberta}$
- learning rate: [5e-6, **1e-5**]
- rumi length: [**3**, 5, 10]$_{roberta}$, [**1**, 3, 5]$_{deberta}$

*Abductive Natural Language Inference (aNLI)* .
- epoch: [**3**, 5, 7]$_{roberta}$, [3, **5**, 7]$_{deberta}$
- batch size: [8, **16**, 32]$_{roberta}$, [8, 32, **64**]$_{deberta}$
- learning rate: [**5e-6**$_{roberta}$, **8e-6**$_{deberta}$]
- rumi length: [5, 10, **15**]$_{roberta}$, [**1**, 3, 5]$_{deberta}$

*HellaSwag* .
- epoch: [1, **3**, 5]$_{roberta}$, [1, **3**, 5]$_{deberta}$
- batch size: [**8**, 16, 32]$_{roberta}$, [8, **16**, 32]$_{deberta}$
- learning rate: [**5e-6**$_{roberta}$, **1e-5**$_{deberta}$]
- rumi length: [3, **5**, 7]$_{roberta}$, [3, **5**, 7]$_{deberta}$

### A.3.2 Baselines

**QA-GNN** (Yasunaga et al., 2021) makes use of the external KG related to the context to enhance the PLMs.

**GreaseLM** (Zhang et al., 2022) also employs the well-known ConceptNet and constructs a deep fusion model to incorporate the text and knowledge graph information.

**Dragon** (Yasunaga et al., 2022) is a deeply joint language-knowledge foundation model pre-trained from text and KG at scale, which achieves strong performance on reasoning about language and knowledge.

**GKP** (Liu et al., 2022b) prepends the knowledge before the question. The original paper concatenates knowledge before each candidate and compute the probability for each candidate. In our setting, to compare with COT, we provide all the candidates and ask the LLM to obtain the final answer.

**COT** (Wei et al., 2022), we use the chain-of-thought provided by previous work (Fu et al., 2023)[3] and utilize the same number of demonstrations for GKP and Knowledge Rumination.

### A.4 Evaluation Metrics

For the commonsense reasoning task, we use *Accuracy* as the evaluation metric. For the GLUE benchmark, we use the same metric in the original paper.

### A.5 Mention Relevance Score

To score the relevance of each mention conditioned on the question context (§3.2), we use the sentence embedding model: all-roberta-large-v1 from SentenceBert (Reimers and Gurevych, 2020) for calculating cosine-similarity.

### A.6 Retrieval Process from Pre-trained Corpus

| Corpus | #Sents | #Dim | #faiss-index |
|---|---|---|---|
| Wikipedia | 2,621,823 | 1,024 | *indexPQ* |
| ConceptNet | 2,543,176 | 1,024 | *indexPQ* |

Table 6: Statistics of Retrieved corpus.

To identify what our generated contextualized representation is similar to, we use the pre-trained corpus Wikipedia (Milne and Witten, 2008) and external knowledge source ConceptNet (Speer et al., 2017) as the retrieval sources (Table 6). For efficient similarity search, we use the 1024-dimensional hidden representations to create a FAISS (Johnson et al., 2021) index and search for top-20 similar triples/samples. By the way, the type of faiss-index is *indexPQ*, which bases on

---

[3] https://github.com/FranxYao/chain-of-thought-hub

a product quantizer. Stored vectors are approximated by PQ codes.

For ConceptNet, we follow KagNet (Lin et al., 2019), which uses sentence template for generating *TRIPLESTRING* like *diamond can be in jewelry store*. Additionally, we add a *'[MASK]'* token at the end of the *TRIPLESTRING* and then feed it as text inputs. For Wikipedia, we retrieve 10000 samples for each sentence based on the prebuilt index in Pyserini (Lin et al., 2021) and then use the original text as inputs.

## A.7  Downstream Evaluation Datasets

We use the following six commonsense reasoning benchmarks for the experiments in the general domain (§4)

**CommonsenseQA (CSQA)** (Talmor et al., 2019) is a 5-way multiple-choice QA task testing commonsense reasoning. The dataset has 12,102 questions. We use the in-house data splits by (Lin et al., 2019).

**OpenbookQA (OBQA)** (Mihaylov et al., 2018) is a 4-way multiple-choice QA task containing elementary science questions. It has 5,957 questions. We use the original data splits in (Mihaylov and Frank, 2018).

**Social Interaction QA (SocialIQA)** (Sap et al., 2019) is a 3-way multiple-choice QA task testing social commonsense reasoning. It has 37K questions. We use the original data splits in (Sap et al., 2019).

**Physical Interaction QA (PIQA)** (Bisk et al., 2020) is a 2-way multiple-choice QA task testing physics reasoning about objects. It has 20K questions. We split the dev set in half to make in-house dev/test sets.

**HellaSwag** (Zellers et al., 2019) is a 4-way multiple-choice task testing grounded commonsense reasoning about events. It has 70K questions. We split the dev set in half to make in-house dev/test sets.

**Abductive Natural Language Inference (aNLI)** (Bhagavatula et al., 2020) is a 2-way multiple-choice task testing abductive commonsense reasoning. It has 170K questions. We use the original data splits in (Bhagavatula et al., 2020).

## A.8  Examples for Different Error Case

In this section, we shows one example for each error type. For each example, we list the knowledge in descending order by the probability score.

It only takes the highest-score knowledge (the knowledge in bold) as rumination information in our experiment.

### A.8.1  Failure to Utilize

**Question:**

What do people typically do while playing guitar?

**Answer:**

Singing

**Knowledge List:**

- **People play guitar while singing.**
- Playing guitar is an activity.
- Playing guitar is a hobby.
- People usually play guitar while singing.
- People play guitar to entertain others

In this example, "People play guitar while singing" has shown the correct answer to the models, but it still remains wrong.

### A.8.2  Ineffective Rumination

**Question:**

What do people aim to do at work?

**Answer:**

Complete Job

**Knowledge List:**

- **People work to earn money.**
- People aim to earn money.
- People aim to get their work done.
- People aim to do their work.
- People aim to do their job well.

In this example, "people work to earn money" has nothing to do with "Complete Job", while the third one in the list, "People aim to get their work done.", conveys the meaning of "Complete Job".

### A.8.3  Incorrect Memory

**Question:**

Where can a human find clothes that aren't pants?

**Answer:**

Dress Shop

**Knowledge List:**

- **A human can find clothes that aren't pants at the beach.**
- Pants are a type of clothing.
- Clothes that aren't pants are dresses and skirts.
- Pants are the most common type of clothing.
- Pants are not the only type of clothing.

In this example, "A human can find clothes that aren't pants at the beach" is the wrong information for the question.

### A.8.4 Missing Information

**Question:**

The freeway had no traffic and few buildings, where is it?

**Answer:**

Countryside

**Knowledge List:**

- **Freeways are usually in cities.**
- Freeways are usually located in urban areas.
- Freeways are in cities.
- Freeways are located in urban areas.
- Freeways are usually in the middle of cities.

In this example, all the knowledge in list indicate that mostly freeways can be found in cities, lacking of the information about the freeway had no traffic and few buildings.

| Task | Knowledge |
|---|---|
| | Input: What do people use to absorb extra ink from a fountain pen?
Knowledge: **A blotter is used to absorb extra ink from a fountain pen.** |
| | Input: What home entertainment equipment requires cable?
Knowledge: **Cable TV is the most common home entertainment equipment that requires cable.** |
| | Input: The fox walked from the city into the forest, what was it looking for?
Knowledge: **Natural habitats are usually away from cities.** |
| CSQA | Input: Google Maps and other highway and street GPS services have replaced what?
Knowledge: **Electronic maps are the modern version of paper atlas.** |
| | Input: Too many people want exotic snakes. The demand is driving what to carry them?
Knowledge: **Some people raise snakes as pets.** |
| | Input: Before getting a divorce, what did the wife feel who was doing all the work?
Knowledge: **Divorce is usually a result of an unhappy marriage.** |

Table 7: The knowledge we used for GKP and Knowledge Rumination on CSQA, derived from Liu et al. (2022b).

| Task | Knowledge |
|---|---|
| | Input: The sun is responsible for?
Knowledge: **The sun is the source of energy for physical cycles on Earth.** |
| | Input: For it to survive, the horse relied on its owner to bring it what?
Knowledge: **An animal requires nutrients for survival.** |
| OBQA | Input: If the Earth revolved around another planet instead of a star, what might it lack?
Knowledge: **The Earth revolving around the Sun causes the seasons to change**. |
| | Input: A bird such as a penguin can survive in arctic weather due to what?
Knowledge: **Thick feathers can be used for keeping warm.** |
| | Input: The gravitational pull between two objects increases as they are
Knowledge: **As the distance from an object decreases, the pull of gravity on that object increases.** |

Table 8: The knowledge we used for GKP and Knowledge Rumination on OBQA.

| Task | Knowledge |
|------|-----------|
| | Input: What will Quinn want to do next? \n (A) Eat messy snacks (B) help out a friend (C) Pick up the dirty clothes \n Quinn wanted to help me clean my room up because it was so messy. 
 Knowledge: **A messy room likely contains dirty clothes.** |
| | Input: What will Aubrey want to do next? \n (A) help Aubrey go back home (B) keep on partying without the mom (C) going on with the mom \n Sashas mom passed out in the middle of the party. Aubrey took Sashas mom to the hospital. 
 Knowledge: **One should attend to their sick family member.** |
| **SocialIQA** | Input: How would Jan feel afterwards? \n (A) scared of losing the cat (B) normal (C) relieved for fixing the problem \n Their cat kept trying to escape out of the window, so Jan placed an obstacle in the way. 
 Knowledge: **One usually has positive emotions after solving a problem.** |
| | Input: How would Sydney feel afterwards? \n (A) affected (B) like they released their tension (C) worse \n Sydney had so much pent up emotion, they burst into tears at work. 
 Knowledge: **Crying can be a catharsis.** |
| | Input: What does Sydney need to do before this? \n (A) be bad at her job (B) do a good job (C) be lazy \n Sydney got a raise and a new promotion. 
 Knowledge: **Pay raise and promotions are usually results of good job performance.** |

Table 9: The knowledge we used for GKP and Knowledge Rumination on SocailIQA, derived from Liu et al. (2022a).