# OpenReview forum: "Knowledge Rumination for Pre-trained Language Models"
_EMNLP/2023/Conference — EMNLP 2023 Main_

### Official Review · Reviewer_urGt · 2023-08-02

**Typos Grammar Style And Presentation Improvements:** Line 111
**Soundness:** 4

**Excitement:**

4: Strong: This paper deepens the understanding of some phenomenon or lowers the barriers to an existing research direction.

**Paper Topic And Main Contributions:**

The authors of this paper believe that the existing pre-training model has mastered a certain knowledge reserve but cannot exert its ability when solving knowledge-intensive tasks and thus propose a new paradigm dubbed Knowledge Rumination. The proposed knowledge rumination is applied to various language models and the experimental results demonstrate the effectiveness of the new paradigm, which proves that the knowledge stored in PLMs can be better exploited to enhance performance.

**Reasons To Accept:**

The proposed Knowledge Rumination can better utilize the knowledge stored in the parameters and have rich scalability. Experimental results demonstrate that the proposed approach can successfully elicit related knowledge for both small and large PLMs.  Analysis indicates that still a large underestimated amount of knowledge can be retrieved from PLM’s model weights, which is a direction worth continuing to explore.

**Reasons To Reject:**

The author holds the view that pre-training models have knowledge potential, but not all pre-training models have the same knowledge reserves. This paper lacks an in-depth analysis of the impact of knowledge reflection on different types of pre-training models.

**Reproducibility:**

4: Could mostly reproduce the results, but there may be some variation because of sample variance or minor variations in their interpretation of the protocol or method.

**Reviewer Confidence:**

1: Not my area, or paper was hard for me to understand. My evaluation is just an educated guess.

---

> ### Author Rebuttal · Authors · 2023-08-28
>
> Thanks for recognizing the value of our work, your comments are highly aligned with our paper. And we hope the following comments could answer your questions.
>
> > The author holds the view that pre-training models have knowledge potential, but not all pre-training models have the same knowledge reserves. This paper lacks an in-depth analysis of the impact of knowledge reflection on different types of pre-training models.
>
> In our pilot experiments, we conducted tests using different pre-training models with various architectures and sizes, including OPT, RoBERTa, and GPT-3. These experiments revealed that many pre-training models indeed possess knowledge potential.
>
> In our main experiments, we specifically evaluated the performance of RoBERTa, DeBETa, and GPT-3. All of these models demonstrated clear benefits from their knowledge reserves, further supporting our claim about the knowledge potential of pre-training models.
>
> Furthermore, as mentioned by reviewer 3JN6, we regrettably did not consider open-source language models like T5 and Llama in our study due to page and time constraints. However, we are actively exploring these models and their potential implications. Thank you for your suggestions.

---

### Official Review · Reviewer_4uX8 · 2023-08-05

**Soundness:** 5

**Excitement:**

4: Strong: This paper deepens the understanding of some phenomenon or lowers the barriers to an existing research direction.

**Paper Topic And Main Contributions:**

This paper proposes the Knowledge Rumination method to help PLMs utilize the latent knowledge they have without retrieving it from an external corpus.
The idea is inspired by the process of rumination in animals, where they regurgitate food to chew it again for better digestion. Similarly, the model is prompted to "review" or "recall" its latent knowledge and then use it for task-specific applications.
A pilot experimental case is presented where a PLM can probe related commonsense knowledge but fails to obtain the correct answer when fine-tuned. This observation motivates the need for knowledge rumination.
To stimulate the model to recall latent knowledge, the authors propose two approaches (1) adding a prompt like "As far as I know" to the PLMs (2) leverage knowledge into the FFN layers of the transformer model explicitly.
The proposed knowledge rumination technique is applied to various PLMs, including RoBERTa, DeBERTa, and GPT-3.
Experimental results on six commonsense reasoning tasks and GLUE benchmarks show the effectiveness of the proposed approach. The results suggest that the knowledge stored in PLMs can be better exploited to enhance performance.

**Questions For The Authors:**

Given the marginal performance improvements observed in some tasks, have you conducted a significance test to validate the statistical significance of these improvements? If not, why?

The application of your method on the GLUE benchmark lacks a clear theoretical explanation, especially since GLUE does not explicitly focus on entity knowledge. Could you provide a more in-depth rationale or justification for the observed performance on GLUE tasks?

How generalizable is the proposed "Knowledge Rumination" approach? Have you tested or considered its applicability across a broader range of NLP tasks beyond the ones presented in the paper?

**Reasons To Accept:**

1. The paper introduces a new and innovative paradigm, "Knowledge Rumination", which is a fresh perspective on how to utilize the latent knowledge within pre-trained language models (PLMs), offering a unique solution to a known problem in the NLP community.

2. The proposed approach is model agnostic， which means it can be applied to any PLMs, making it versatile and widely applicable across various models and architectures.

3. The paper provides comprehensive experimental results on multiple commonsense reasoning tasks and GLUE benchmarks.

4. The approach, while conceptually novel, is straightforward to implement. The idea of using task-guided prompting, like "As far as I know", is simple yet shows significant improvements in model performance.

**Reasons To Reject:**

1. The performance improvements achieved by the proposed method might not be very significant. It's crucial for the authors to conduct a significance test to validate whether the observed improvements are statistically significant or just due to random variations.

2. The performance improvements achieved by the proposed method might not be very significant. It's crucial for the authors to conduct a significance test to validate whether the observed improvements are statistically significant or just due to random variations.

3. The approach heavily relies on task-guided prompting, like "As far as I know". While this is innovative, the effectiveness of such prompts might vary across tasks, and there might be a need for manual tuning or customization for different applications.

**Reproducibility:**

4: Could mostly reproduce the results, but there may be some variation because of sample variance or minor variations in their interpretation of the protocol or method.

**Reviewer Confidence:**

4: Quite sure. I tried to check the important points carefully. It's unlikely, though conceivable, that I missed something that should affect my ratings.

---

> ### Author Rebuttal · Authors · 2023-08-28
>
> Thanks for your valuable reviews. We have made the following revisions to address your concerns.
> > The performance improvements achieved by the proposed method might not be very significant. It's crucial for the authors to conduct a significance test to validate whether the observed improvements are statistically significant or just due to random variations.
>
> Thank you for the reminder. We have conducted a significance test to address this concern. We computed the 95% confidence intervals and performed a t-test. The results are as follows:
> | Model      | CSQA | SocialIQA
> | ----------- | ----------- | -----------
> | RoBerta      |   67.03($\pm 1.2$)     | 74.2($\pm 1.5$)
> | RumiRoBERTa   |   70.3($\pm 0.6$)      | 77.3 ($\pm 0.5$)
>
> For SocialIQA,
> - t statistic: -5.485490552046735
> - **p value**: 0.0005836574037182736
>
> For CSQA,
> - t statistic: -5.962493710766469
> - **p value**: 0.00033722640767272
>
> Both p-values are less than 0.005, indicating that the improvements achieved by our method are statistically significant. We will include this information in our table and conduct further tests on additional datasets.
>
> > The application of your method on the GLUE benchmark lacks a clear theoretical explanation, especially since GLUE does not explicitly focus on entity knowledge. Could you provide a more in-depth rationale or justification for the observed performance on GLUE tasks?
>
> While most research currently focuses on entity or factual knowledge stored in pre-trained language models (PLMs), it is important to note that PLMs also store other types of knowledge, including skill knowledge and task knowledge. This was demonstrated in Xiaozhi Wang's paper titled "Finding Skill Neurons in Pre-trained Transformer-based Language Models," presented at EMNLP 2022.
>
> In our paper, we designed the task prompt to encourage the model to reflect on the skill knowledge encoded within its parameters. Based on the observed performance on the GLUE tasks, we conclude that PLMs can indeed benefit from this knowledge. We will provide a more detailed explanation and justification in our paper.
>
> > How generalizable is the proposed "Knowledge Rumination" approach? Have you tested or considered its applicability across a broader range of NLP tasks beyond the ones presented in the paper?
>
> While our paper focused on six commonsense QA tasks and the GLUE benchmark, which cover a wide range of NLP tasks, we acknowledge that there are other tasks that were not specifically tested in our experiments. We believe that our method has the potential for application across a broader range of NLP tasks and plan to conduct more experiments to explore its effectiveness in those domains.

---

### Official Review · Reviewer_6u65 · 2023-08-09

**Soundness:** 4

**Excitement:**

4: Strong: This paper deepens the understanding of some phenomenon or lowers the barriers to an existing research direction.

**Paper Topic And Main Contributions:**

The paper aims to improve the performance of Pre-trained Language Models (PLMs) on knowledge-intensive tasks without making use of external knowledge sources. To that end, the paper proposes to first query PLMs with the input (question) and some unique prompts to help PLMs to “reflect” on their knowledge of the question, and then provide the output (in latent space) as an additional input to help PLMs answer the initial question.
The proposed method is evaluated on several commonsense reasoning QA, and is shown to outeperform baselines the depend on external knowledge. An analysis is conducted to investigate the choice of injecting the additional knowledge in the last layer as opposed to the first layer (after the embeddings layer), and to look into how the proposed method help PLMs elicit more knowledge. Additionally, an evaluation is conducted on a large language model (GPT-3), where the method is shown to outperform other baselines on 2 out of 3 datasets.

**Questions For The Authors:**

Question 6u65_A : Have you investigated which types of prompts contribute the most to the improved performance? The top 1 mention is an interesting and simple baseline.

Question 6u65_B : in Table 3, does "Finetuning" refer to direct finetuning on the source or the target dataset?

**Reasons To Accept:**

* The paper proposes a novel method to improve the performance of PLMs on knowledge-intensive task by “re-using” knowledge that comes from the PLMs themselves
* The introduced method outperform baselines that depend on external knowledge
* The conducted evaluations verifies the efficacy of the method


**Reasons To Reject:**

NA

**Reproducibility:**

4: Could mostly reproduce the results, but there may be some variation because of sample variance or minor variations in their interpretation of the protocol or method.

**Reviewer Confidence:**

3: Pretty sure, but there's a chance I missed something. Although I have a good feel for this area in general, I did not carefully check the paper's details, e.g., the math, experimental design, or novelty.

**Typos Grammar Style And Presentation Improvements:**

L84: absoprtionby
L86: chewingwe
L94: FFN: add "Feed Forward Network"

---

> ### Author Rebuttal · Authors · 2023-08-28
>
> Thanks for acknowledging the value of our work. We hope the following comments address your questions.
> > Have you investigated which types of prompts contribute the most to the improved performance? The top 1 mention is an interesting and simple baseline.
>
> Yes, we have investigated the impact of different types of prompts on performance. For GLUE tasks, the 'task prompt' is found to be more influential, while for commonsense QA tasks, background information plays a more crucial role. We will include this information in our paper.
>
> During our experiments, we initially filtered the mentions using a threshold and selected the top-2 mentions. However, we observed that in most cases, only one mention's score surpassed the threshold, indicating the significance of that particular mention.
>
> > In Table 3, does "Finetuning" refer to direct finetuning on the source or the target dataset?
>
> Yes, the finetuning refers to direct finetuning on the source datasets and test on the target datasets. Rumination is similar, here, we train the model to ruminate the source datasets and test on the target dataset.

---

### Official Review · Reviewer_3JN6 · 2023-08-10

**Soundness:** 4

**Excitement:**

4: Strong: This paper deepens the understanding of some phenomenon or lowers the barriers to an existing research direction.

**Missing References:**

Zhou, Denny, et al. "Least-to-most prompting enables complex reasoning in large language models." arXiv preprint arXiv:2205.10625 (2022).

**Paper Topic And Main Contributions:**

The paper introduces Knowledge Rumination, a technique to extract and utilize latent knowledge from pre-trained models for knowledge-intensive tasks. It considers the hidden states of well-designed prompts as knowledge sources and integrates them into the hidden layers of task-specific models through end-to-end fine-tuning. The paper shows that the technique achieves promising results on various tasks. I appreciate the direction of this work as it leverages the knowledge already embedded in pre-trained weights instead of relying on an external knowledge base, which can be costly. However, I also observe that the prompt design is similar to the least-to-most prompting approach proposed by Zhou et al. (2022), which also decomposes a problem into multiple steps for easier solving. I suggest that the authors should cite and compare with this related work. Moreover, I find that the application of the technique to GPT-3.5 is not well-motivated as this closed-source model does not output soft hidden states as knowledge but discrete tokens that prevent back-propagation. I think this is a limitation and I recommend that the authors should either conduct experiments on some open-sourced causal LMs such as OPT or LLaMA, or develop methods that can handle black-box LLMs.

**Reasons To Accept:**

1. An interesting method to elicit and exploit latent knowledge of pre-trained models to solve knowledge-intensive tasks.
2. Extensive experiments show promising results.

**Reasons To Reject:**

1. The application on GPT-3.5 is not well-motivated.
2. Missing citations.

**Reproducibility:**

4: Could mostly reproduce the results, but there may be some variation because of sample variance or minor variations in their interpretation of the protocol or method.

**Reviewer Confidence:**

3: Pretty sure, but there's a chance I missed something. Although I have a good feel for this area in general, I did not carefully check the paper's details, e.g., the math, experimental design, or novelty.

---

> ### Author Rebuttal · Authors · 2023-08-28
>
> Thank you for your valuable suggestions, and we will ensure to add the missing citations to our paper.
>
> **Our work differs from the 'least-to-most prompt' approach**.
> Firstly, our method does not involve decomposing the task.
> Instead, our prompt aims to encourage the model to reconsider the knowledge it has learned about the task and the associated question.
> This knowledge encompasses background information and question-specific commonsense. In contrast, the 'least-to-most prompt' decomposes the question into sub-questions and asks the model to answer each sub-question.
> Moreover, **even when answering the sub-questions, the 'least-to-most prompt' model still faces the problem mentioned in our paper where it may have learned the answer to the sub-question within its parameters but fails to provide the correct answer**.  Our knowledge rumination approach is orthogonal to the 'least-to-most prompt' approach'. Therefore, combining rumination with the 'least-to-most prompt' approach may lead to a stronger solution.
>
> Sorry for the unclear motivation of GPT-3.5. Our experiments demonstrate that GPT-3.5 also experiences the phenomena presented in our paper. We aim to transfer our methods to large language models (LLMs), and we have modified our method to suit LLMs. Detailed discussions on this modification are provided in Section 5.3. Our approach is applicable to black-box models, and its effectiveness has been demonstrated in comparison to COT. Additionally, the 'least-to-most prompt' approach is also applied to black-box models, and our method can be combined with it.
>
> We will try to consider knowledge rumination with open-sourced causal LM models like GPT and LLaMA. Thank you for your great suggestions!

---

### Meta-Review · Area_Chair_HyXM · 2023-09-19

**Recommendation:** 4

**Metareview:**

This paper introduces a novel technique (coined Knowledge Rumination) aimed at improving the performance of Pre-trained Language Models (PLMs) on knowledge-intensive tasks without relying on external knowledge sources. The approach elicits latent knowledge already embedded in PLMs via task-guided prompting and injects it back into the feed-forward network for knowledge consolidation. The paper presents experiments on various PLMs, including RoBERTa, DeBERTa, and GPT-3, showing improved performance on commonsense reasoning tasks and benchmark datasets. This highlights the potential for better exploiting the knowledge already present within PLMs to enhance their performance.

Although the concept of extracting knowledge to enhance a PLM's performance, whether from external knowledge bases, external language models, or even the model itself, is not novel, this paper introduces a unique, innovative method to elicit knowledge already present in a PLM and to consolidate it to better address commonsense reasoning tasks.

Overall, the authors conduct comprehensive experiments on multiple commonsense reasoning tasks and GLUE benchmarks, showcasing the effectiveness of the approach. The primary limitation might be attributed to the relatively modest performance improvement achieved by the proposed approach, even though this improvement is fairly consistent across all tasks and language models.

---

### Decision · Program_Chairs · 2023-10-07

**Decision:**

Accept-Main

**Comment:**

This paper introduces a novel technique (coined Knowledge Rumination) aimed at improving the performance of Pre-trained Language Models (PLMs) on knowledge-intensive tasks without relying on external knowledge sources. The approach elicits latent knowledge already embedded in PLMs via task-guided prompting and injects it back into the feed-forward network for knowledge consolidation. The paper presents experiments on various PLMs, including RoBERTa, DeBERTa, and GPT-3, showing improved performance on commonsense reasoning tasks and benchmark datasets. This highlights the potential for better exploiting the knowledge already present within PLMs to enhance their performance.

Although the concept of extracting knowledge to enhance a PLM's performance, whether from external knowledge bases, external language models, or even the model itself, is not novel, this paper introduces a unique, innovative method to elicit knowledge already present in a PLM and to consolidate it to better address commonsense reasoning tasks.

Overall, the authors conduct comprehensive experiments on multiple commonsense reasoning tasks and GLUE benchmarks, showcasing the effectiveness of the approach. The primary limitation might be attributed to the relatively modest performance improvement achieved by the proposed approach, even though this improvement is fairly consistent across all tasks and language models.